# Population Genetic Structure of a Rare Butterfly in a Fragmented South Florida Ecosystem

**DOI:** 10.3390/insects14040321

**Published:** 2023-03-27

**Authors:** Emily Heffernan, Amanda Markee, Mary R. Truglio, Megan Barkdull, Sarah Steele Cabrera, Jaret Daniels

**Affiliations:** 1McGuire Center for Lepidoptera and Biodiversity, Florida Museum of Natural History, 3215 Hull Road, Gainesville, FL 32611, USA; 2New College of Florida, 5800 Bay Shore Road, Sarasota, FL 34243, USA; 3Entomology and Nematology Department, University of Florida, Steinmetz Hall, 1881 Natural Area Dr., Gainesville, FL 32611, USA; 4Department of Ecology and Evolutionary Biology, Cornell University, 215 Tower Rd, Ithaca, NY 14853, USA

**Keywords:** *Ephyriades brunnea floridensis*, pine rockland, microsatellites, *Wolbachia*, gene flow, genetic diversity, population structure

## Abstract

**Simple Summary:**

Biodiversity loss at both species and genetic scales has been exacerbated in recent decades due to habitat fragmentation and destruction. These losses have been especially devastating to insects. Butterflies are charismatic microfauna that offer an excellent opportunity to evaluate the consequences of how fragmentation disrupts gene flow between population segments and may serve as indicators for other less visible insect species. The Florida duskywing butterfly is a rare species that is dependent on a critically endangered habitat, the Florida pine rocklands. We ran a series of genetic studies, akin to DNA fingerprinting, to assess the genetic health of the Florida duskywing as well as to evaluate the level of connectivity between the remaining populations. Despite declining population sizes, we found that populations still hold a unique genetic diversity. Genetic results also showed that these butterflies could be clustered into mainland and Florida Keys population groups, which were connected by a moderate gene flow. Additionally, no sampled individuals tested positive for a bacterial infection that has harmed other butterfly populations, meaning that individuals can be moved between populations without major concern for bacterial spread. In summary, this work supports translocations of individuals between populations to (1) facilitate an enhanced gene flow and (2) reinforce small population sizes.

**Abstract:**

We investigated the genetic structure and diversity between populations of a rare butterfly, the Florida duskywing (*Ephyriades brunnea floridensis* E. Bell and W. Comstock, 1948) (Lepidoptera: Hesperiidae) across a network of South Florida pine rockland habitat fragments. Based on 81 individuals from seven populations and using multiple polymorphic microsatellite loci, our analyses support the presence of mainland Florida (peninsular) and Florida Keys (island) population groupings, with a moderate, asymmetrical gene flow connecting them, and the presence of private alleles providing unique identities to each. We additionally found that despite a prevalence in many Lepidoptera, the presence of *Wolbachia* was not identified in any of the samples screened. Our findings can be used to inform conservation and recovery decisions, including population monitoring, organism translocation, and priority areas for management, restoration or stepping-stone creation to help maintain the complex genetic structure of separate populations.

## 1. Introduction

A multitude of anthropogenic drivers continue to rapidly erode and threaten global biodiversity [1,2]. Although losses are better documented in more well-studied taxa, a multitude of recent studies continue to chronicle insect declines [3,4]. Of the many stressors, land-use change, particularly habitat alteration, loss, and fragmentation, are among the leading causes of population declines and isolation [5]. Small populations are particularly at risk of extinction due to a decreased organism density and abundance, ecological processes such as environmental and demographic stochasticity, and the alteration of the dispersal and gene flow, resulting in a genetic compromise by inbreeding, drift and mutational meltdown (Figure 1) [6,7,8,9]. Such impacts may be particularly severe for species with narrow ecological niches, e.g., specialists [10]. Ensuring population connectivity is essential for species’ persistence and resilience in fragmented landscapes, and a key target in conservation. Equally informative for species management and recovery planning is an understanding of the current population structure and identification of landscape features that restrict or otherwise offer barriers to gene flow.

In the current study, we targeted a network of remnant pine rockland habitat fragments in South Florida in the United States as a model system that was ideal to investigate functional linkages between spatially structured populations. Pine rockland is a globally critically imperiled ecosystem that occurs only in South Florida and the Caribbean [12]. Within South Florida, the pine rockland habitat is confined primarily to the Miami Rock Ridge, extending from Miami-Dade County southwest to Long Pine Key in Everglades National Park, as well as scattered fragments in the Florida Keys and Big Cypress National Preserve. Extensive urbanization, dominated by the rapid expansion of the greater Miami metropolitan area over the past century, severely reduced the extent of this upland habitat in South Florida. Historically occupying over 750,000 ha prior to European settlement, less than 5% of that total remains today, with the largest contiguous parcel of some 8029 ha existing within Everglades National Park. Within the broader urban matrix of Miami-Dade County, intact pine rockland parcels are severely fragmented and isolated, with most only a few hectares in size (Figure 2).

Pine rocklands are a unique fire-adapted community that supports a rich diversity of temperate and tropical species, including many endemic plant and animal taxa, several listed under the U.S. Endangered Species Act (ESA). Of these, three insects, the Florida leafwing (*Anaea troglodyta floridalis*), Bartram’s scrub-hairstreak (*Strymon acis bartrami*), and Miami tiger beetle (*Cicindelidia floridana*) are all endemic to Florida (USA), restricted entirely to the pine rockland habitat, and listed as Endangered under the ESA. A fourth pine rockland specialist, the Florida duskywing (*Ephyriades brunnea floridensis*) (Figure 3), is designated as a Species of Greatest Conservation Need (SGCN) in Florida’s State Wildlife Action Plan [13]. This rare butterfly was also among the top three highest-priority ranked species based on a comprehensive statewide combined vulnerability assessment and conservation prioritization [14].

While habitat fragmentation alone can pose the risk of extirpation for small populations, pathogens like *Wolbachia* can synergistically increase this risk. *Wolbachia* are a diverse group of intracellular bacteria with multiple host adaptations. When *Wolbachia* acts as a reproductive parasite, it can cause reproductive fitness advantages in infected females [15]. Altered sex ratios caused by the reproductive phenotypes of *Wolbachia* can create demographic challenges, especially in small populations. When considering conservation and management strategies, such as translocation and assisted breeding programs, *Wolbachia* screening should be taken into consideration to decrease the potential for pathogen transmission. 

Despite this ranking, *E. b. floridensis* can at times be locally common, particularly in larger parcels across the existing South Florida pine rockland habitat network. Due to its relative abundance in comparison with other pine rockland specialists, *E. b. floridensis* represents an appropriate surrogate for other rare insects with similarly limited or presumedly limited dispersal abilities. Therefore, we conducted a multi-population assessment of the *E. b. floridensis* to (i) evaluate the genetic diversity, population genetic structure and gene flow at the landscape and regional levels, (ii) screen for the presence of *Wolbachia* infection, and (iii) identify priority conservation actions to help maintain or enhance network maintenance and gene flow.

## 2. Materials and Methods

### 2.1. Study Area

We utilized a fragmented network of South Florida pine rockland habitat parcels in Miami-Dade and Monroe Counties, Florida (USA) as our study system. This included a total of 14 conservation properties that ranged in size from <6.0 ha to over 5000 ha and of various ownership and management oversights, including county, state and federal agencies (Figure 2).

### 2.2. Site Selection

Properties targeted for sampling were selected based on several criteria. These included: (i) a documented presence of *E. b. floridensis* over the past two years; (ii) a location to sample the broadest possible geographic range within Florida; and (iii) the ability to secure appropriate property access and permits for sampling.

### 2.3. Sample Collection and DNA Extraction

We sampled a total of 81 individuals of *E. b. floridensis* from a total of 7 of 14 conservation property locations over a four-year period from December 2016 to September 2019 for genetic studies (Figure 2). For those conservation properties lacking tissue samples, no *E. b. floridensis* were recorded during our field surveys. As all conservation properties sampled were spatially discrete, surrounded by a considerable urban matrix or other habitat types, or existed on individual islands within the Florida Keys, we proceeded with the assumption that they supported discrete populations. Trained field technicians nonlethally collected tissue samples from adult legs, following protocols of [16,17,18], preserved samples in 1.5 mL microcentrifuge tubes filled with 90–100% ethanol, and stored all samples in −80 °C freezers to prevent degradation until DNA extraction. We extracted genomic DNA using a modified Qiagen^®^ DNeasy Blood and Tissue Kit (Valencia, CA, USA) extraction protocol and quantified genomic DNA concentrations using a Nanodrop spectrophotometer to assess the quality and quantity of each sample. For whole genome sequencing and marker development, we extracted DNA from the legs and thorax of a voucher specimen (a male collected from Big Pine Key Florida, Award Number F17AP00467) and treated the sample with 4 µL RNAse. We used a GenomiphiTM V2 DNA AMplification Kit to amplify genomic DNA and sequenced the whole genome on a PacBio RSII at the University of Florida Interdisciplinary Center for Biotechnology Research (ICBR). Genomic DNA was sheared into 1 kb fragments with an ultrasonicator and prepared for sequencing by annealing adaptors (Pacific Biosciences, Menlo Park, CA, USA). We used one SMRT cell for the PacBio RSII sequencing run, completed in the circular consensus mode. We used a Globus Personal Connect client with UF-ICBR cyberinfrastructure to receive the consensus PacBio RSII data.

### 2.4. Microsatellite Marker Development and Analyses

We formatted input files (from over 3 billion nucleotides read) and ran them through Geneious (using the Phobos extension) (Mayer, Christoph, Phobos 3.3.11, 2006–2010) to identify microsatellite markers. We interfaced with BatchPrimer 3 [19] to develop primers to amplify microsatellite markers. We compiled an extensive database of thousands of potential loci and primers and began testing based on the motif type and length as well as low penalty scores. We added a known sequence M13 tag to the 5′ end of each forward primer to allow for fluorescent probes to be added during the PCR step to evaluate the primers, following the protocols of [20]. We tested a total of 24 microsatellite loci. To verify that these microsatellite-containing reads were (1) butterfly DNA and (2) not linked to coding genes (an initial screen to verify that they are not under selection and are neutral), we compared reads against the NCBI GenBank database via BLAST [21,22].

### 2.5. PCR Processing

We followed the reaction conditions and PCR concentrations of Saarinen and Austin (2010). This protocol uses M13 primer labeling and fluorescently labeled tags and allows for multi-pooling of PCR products. PCR products were diluted 100× and multipooled for fragment size analysis on an ABI 3730 sequencer (University of Florida, Gainesville, FL, USA). We performed all genotyping analyses with GeneMarker v. 2.6.3 (SoftGenetics, LLC; State College, PA, USA) [23], and all allele sizes were manually confirmed.

### 2.6. Statistical Analyses

We calculated the linkage disequilibrium (LD) between loci and exact tests of the Hardy–Weinberg equilibrium (HWE) in Arelquin v.3.5.2.2 [24]. Sequential Bonferroni corrections were used to control for multiple comparisons for both the LD and HWE datasets, respectively [25].

We calculated summary statistics, observed and expected heterozygosity, and FIS (inbreeding coefficient) in each population using GenAlEx 6.5 [26]. To quantify genetic differentiation between samples, we calculated pairwise FST values as well G’ST [26] using GenAlEx 6.5, each permuted 999 times. G’ST was used to account for the highly variable nature of microsatellites and to correct for bias by normalizing FST. We performed an analysis of molecular variance (AMOVA) to evaluate where genetic variance was the greatest (between individuals or between populations).

We used two complementary methods to evaluate genetic structuring. First, we ran principal coordinates analyses (PCoA) of genetic distances between individuals using GenAlEx 6.5 [27] for a visual representation of the genetic distance. Next, we used Structure 2.3.4 [28] to evaluate how many discrete populations (K values) were supported in the data. We ran the admixture model with correlated allele frequencies for each value of K (for K = 1 to K = 5). Each value of K was run 20 times, using a parameter set of 500,000 burn-in and 750,000 MCMC replications after the burn-in. We used the program Structure Harvester [29] to calculate the average L(K) and standard deviation across the 20 runs for each K and then calculate the ln Pr(X|K) using the ad hoc delta K value [30] to help infer which K was the most likely where different K values had similar likelihoods. We used the program CLUMPP [31] to combine and Distruct [32] to visually represent the results of the Structure analyses and distribution of K population groups.

### 2.7. Wolbachia Screening

Here, we screened all 81 individuals of *E. b. floridensis* for *Wolbachia*. We extracted genomic DNA from all individuals using a modified Qiagen^®^ DNeasy Blood and Tissue Kit (Valencia, CA, USA) extraction protocol and quantified genomic DNA concentrations using a Nanodrop spectrophotometer. To screen for *Wolbachia*, we performed two PCR reactions following the methods outlined in [33]: one reaction using primers to amplify *Wolbachia* 16S rDNA (WSpecF and WSpecR), and another reaction using universal arthropod primers for 28S rDNA to confirm negative *Wolbachia* results (28sF3633 and 28sR4076). We followed the PCR cycling conditions described in [33] and concurrently ran standard positive and negative controls for each *Wolbachia* screening. We visualized PCR amplification products on 1% agarose gels to determine the presence of *Wolbachia*.

## 3. Results

### 3.1. Microsatellite Marker Development and Analyses 

The results from the PacBio RSII run yielded 3,405,242,620 nucleotide bases read with an average read length of 27,385 bases. Of the 24 microsatellite loci we began testing, 18 had perfect repeats and no linkage to coding regions. Of these 18 loci, 5 loci consistently amplified across all templates and lacked large allele dropout and stutter problems. These 5 loci are highly informative, with 4–16 alleles per loci (Appendix A). After the Bonferroni correction, all microsatellite loci were randomly associated and in linkage equilibrium in all populations, and summary statistics are provided (Table 1).

Individual populations are not in Hardy−Weinberg equilibrium at all loci (*p* > 0.05 after Bonferroni correction), although the two largest populations (Big Pine Key and Zoo Miami) are in Hardy−Weinberg equilibrium at 3/5 and 4/5 loci, respectively.

A summary of pairwise FST values indicates a moderate gene flow within the mainland and island groups, respectively (e.g., FST = 0.111 between Big Pine Key and Dagny Johnson; FST = 0.079 between Everglades National Park and Zoo Miami).

FST values also show similar signs of moderate connectivity between the two major groups (FST = 0.083 between the mainland and island). These values underscore the similarity between the mainland and island groups and are helpful in interpreting the next set of analyses showing two groups. This can be visualized by the PCoA analysis of samples (Figure 4). The AMOVA results provide additional support, with 64% and 24% of variation within and among individuals (respectively) and only 12% variation among different populations.

Multiple analyses (a principal coordinate analysis and a Bayesian clustering analysis) confirm the general presence of two-three population groupings (Table 2, Figure 4 and Figure 5). These groups assemble into a mainland group of four population samples (Everglades National Park, Zoo Miami, Eachus Pineland, and Coast Guard). Geographic coordinates of samples confirm that the Eachus Pineland and Coast Guard areas are adjacent to Zoo Miami. The Nixon Smiley set of samples (N = 3) maps geographically near Zoo Miami but does not match with either the mainland or island population groupings in the genetic PCoA analysis. The small sample size in multiple populations is likely the cause of the three population groupings, whereas most evidence supports a mainland and island population.

The island group consists of the Big Pine Key and Dagny Johnson samples. Big Pine Key is an island in the lower Florida Keys and is ~141 km south of Dagny Johnson Hammock State Park on the upper keys island of Key Largo.

### 3.2. Wolbachia Screening Results

Despite the geographic isolation and small population sizes, we did not detect the presence of *Wolbachia* in any of the 81 individuals of *E. b. floridensis*. Standard genomic DNA positive controls were amplified on all 1% agarose visualizations, and standard negative control amplification was absent. We verified PCR viability for all genomic DNA using 28S universal arthropod primers, and we found all 81 samples amplified at the appropriate size. *Wolbachia* may cause changes in the population structure, inferred by mitochondrial signatures (as described by [33] in the Karner blue butterfly and by [34] in Palaearctic blue butterflies). In view of the fact that our study used microsatellite markers (derived from nuclear DNA, not mitochondrial DNA) and did not detect any Wolbachia, we have increased confidence in our results on the population structure.

## 4. Discussion

We observed a considerable number of private alleles in the larger populations (mainland Zoo Miami and island Big Pine Key), supporting our original hypothesis that a unique diversity persists in these populations. These also represent populations at or near the northernmost and southernmost geographic extent of *E. b. floridensis*. The fact that so many private alleles remain in the larger populations is positive and suggests that a significant genetic drift has not yet occurred. These populations therefore warrant increased protection and management to prevent the erosion of existing genetic diversity. Specifically, a strategy to maintain genetic diversity via captive breeding or translocation programs to safeguard against loss is supported by these data.

Small populations are susceptible to extirpation by stochastic events. We support the continued monitoring of all small populations and, as necessary, support the translocation of individuals from other sites to augment a small population size, artificially maintain gene flow, and overcome effects of genetic drift.

Our results also support the hypothesis that an asymmetrical gene flow connects remaining populations in mainland Florida and the southern Florida Keys. At present, we observe a moderate connectivity between the mainland and Keys populations, with similar allele frequencies uniting the groups and the presence of private alleles providing unique identities to each. Therefore, we support activities that maintain the complex genetic structure of separate populations, each maintaining a unique genetic diversity and moderate levels of connectivity. We also advocate sampling additional populations to help establish a detailed picture of the metapopulation arrangement of the Florida Duskywing across its range. Similarly, as the extent and configuration of the occupied habitat, *E. b. floridensis* population abundance, host density, or connectivity measures were considered to be outside of the current project scope and not determined for each conservation property, addressing these metrics remains a focus for future research, in order to better understand the ecology and metapopulation dynamics of this increasingly at-risk butterfly.

## 5. Conclusions

In summary, this work supports translocations of individuals between populations in order to (1) support a discontinued gene flow and (2) reinforce small population sizes. A preliminary screening of all individuals, following [33] methods, demonstrate no *Wolbachia* infection in Florida Duskywing butterflies. As *Wolbachia* infection may have unintended consequences for populations (e.g., male-killing, etc.) we advocate for populations to continue to be screened before any translocation or captive breeding involving the mixing of multiple populations occurs. The negative *Wolbachia* results found in this work allow for captive breeding, head-starting, and translocation to remain viable conservation strategies. Given the multiple threats facing this species, including increased hurricane activity, we consider that the continued monitoring of connectivity and maintenance of specific populations enhance the security of this taxon.

## Figures and Tables

**Figure 1 insects-14-00321-f001:**
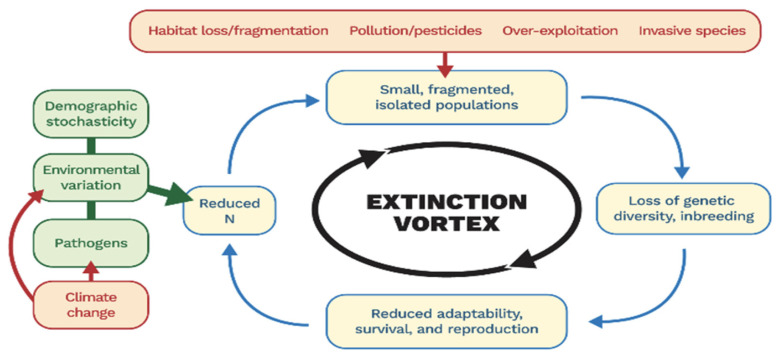
Multiple synergistic threats (red) exacerbate natural variation (green) and reduce diversity (gold) to perpetuate species extinction vortex (redrawn and edited from [11] Figure 2.3).

**Figure 2 insects-14-00321-f002:**
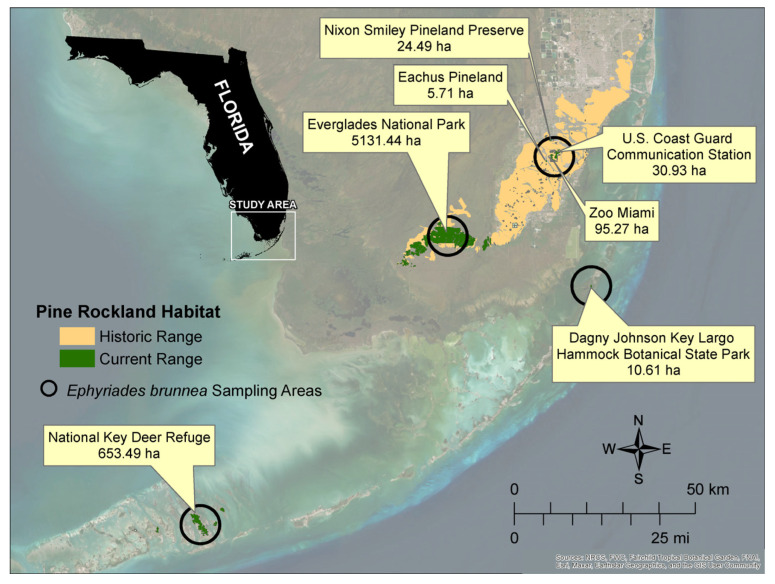
Map showing historic and current range of pine rockland habitat in South Florida along with *E. b. floridensis* conservation property sampling areas extending from the Lower Florida Keys north to Miami-Dade County. Properties where *E. b. floridensis* samples were taken are noted, as well as the current size of pine rockland habitat within that property. Areas were calculated using Florida Cooperative Land Cover Map developed by FWC and FNAI (version 3.6).

**Figure 3 insects-14-00321-f003:**
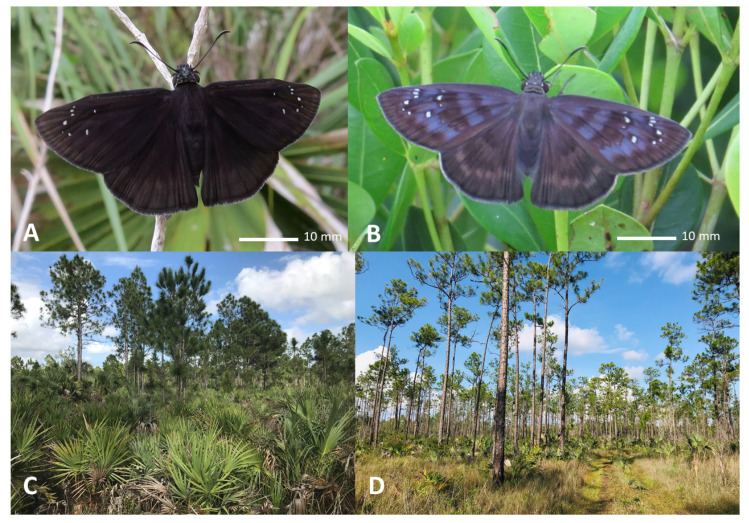
Adult male Florida duskywing (*E. brunnea floridensis*): (**A**) adult female Florida duskywing (*E. b. floridensis*); (**B**) Florida duskywing (*E. b. floridensis*): (**C**) pine rockland habitat at Zoo Miami (Miami-Dade County, Florida, USA); and (**D**) pine rockand habitat in Evergaldes National Park (Miami-Dade County, Florida, USA).

**Figure 4 insects-14-00321-f004:**
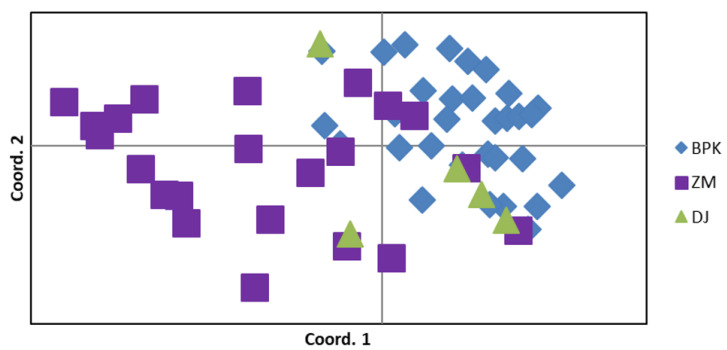
Principle coordinate analysis (BPK= Big Pine Key; ZM = Zoo Miami; DJ = Dagny Johnson) showing two populations connected by genetic similarity.

**Figure 5 insects-14-00321-f005:**
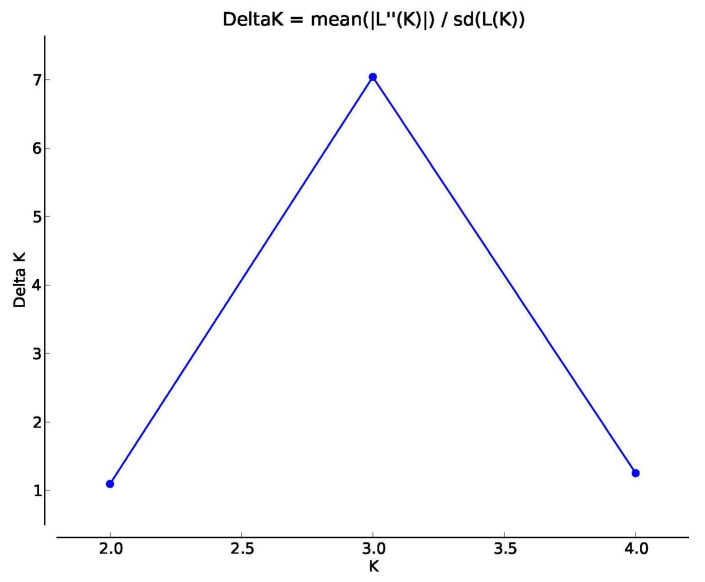
Results of the Bayesian Structure analyses and distribution of K = 3 population groups.

**Table 1 insects-14-00321-t001:** Population-level statistics for *E. b. floridensis* from 7 population locations (conservation properties) in South Florida. N, sample size; Na, number of different alleles; Ne, number of effective alleles; Ho, observed heterozygosity; He, expected heterozygosity; F, Fixation Index = (He − Ho)/He = 1 − (Ho/He); BPK, Big Pine Key (within National Key Deer Refuge); ENP, Everglades National Park. Big Pine Key and Dagny Johnson are island populations; all other populations exist on the Florida mainland.

Population	N	Na	Ne	Ho	He	F
BPK	31	6.600	4.604	0.497	0.660	0.254
ZooMiami	22	5.200	3.949	0.523	0.608	0.169
DagnyJohnson	5	1.800	1.557	0.160	0.320	0.583
NixonSmiley	3	1.600	1.477	0.467	0.256	−0.733
CoastGuard	7	2.800	2.234	0.373	0.528	0.345
EachusPineland	3	2.600	2.357	0.467	0.522	0.068
ENP	10	3.400	2.534	0.582	0.513	−0.142

**Table 2 insects-14-00321-t002:** Structure results of population clustering.

K	Mean LnP(K)	Stdev LnP(K)	Ln′(K)	[Ln″(K)]	DeltaK
1	−792.850	0.416	–	–	–
2	−832.895	57.481	−40.045	62.955	1.095
3	−809.985	12.901	22.910	90.845	7.042
4	−877.920	47.633	−67.935	59.680	1.253
5	−886.175	24.719	−8.255	–	–

## Data Availability

The data presented in this study are available at the Sequence Read Archive (SRA), BioProject accession number PRJNA932780.

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
