# Peer review of "Population Genetic Structure of a Rare Butterfly in a Fragmented South Florida Ecosystem"

_insects, 2023, doi:10.3390/insects14040321_

Round 1
Reviewer 1 Report
The manuscript “Population Genetic Structure of a Rare Butterfly in a Fragmented South Florida Ecosystem” authored by Emily Heffernan et al. represents an excellent case study in population genetic health of a skipper species under anthropogenic disturbance. The methodology, results, and discussion of this manuscript are all well organised and written. Here I only have a few minor comments. I would like to recommend a quick publication of this research.
Minor comments:
1. Figure 2 is a spherical aspect of the Florida Keys, which is unfamiliar with most readers outside the United States. Please indicate on the Florida map (the black one), the large black patch is Florida and the rectangle represents the enlarged research area: the Florida Keys. Also, please make the circles more obvious on the image.
2. The A and B annotation on Figure 3 is too small, please enlarge them. Adding a 10 mm scale to each sub-image is better.
3. Figure 4, please remove the outline of this plot, and reduce the size of the symbols.
4. Please consider including the gel images of Wolbachia detection as supplementary material.
Author Response
Thank you for your kind review of our manuscript. Based on your comments, please find the following response to aid in clarification.
Reviewer 1 Comments:
- Figure 2 is a spherical aspect of the Florida Keys, which is unfamiliar with most readers outside the United States. Please indicate on the Florida map (the black one), the large black patch is Florida and the rectangle represents the enlarged research area: the Florida Keys. Also, please make the circles more obvious on the image.
We have modified the figure to more clearly identify the black element as the state of Florida and the rectangle as the research area. We have additionally made the circle (sampling areas) more obvious and added callouts to each sampling property along with the area in hectares of each property.
- The A and B annotation on Figure 3 is too small, please enlarge them. Adding a 10 mm scale to each sub-image is better.
We enlarged the A and B annotation and added a 10 mm scale bar to both images. We additionally added two images of the habitat.
- Figure 4, please remove the outline of this plot, and reduce the size of the symbols.
Outline/border has been removed.
- Please consider including the gel images of Wolbachia detection as supplementary material.
As Wolbachia was not detected in any of our samples screened, including an image(s) of blank gels seems superfluous. Thus, we chose not to include this information in the supplementary materials.
Reviewer 2 Report
This is highly interesting study of population genetic structure of an endangered butterfly (sub)species, using originally designed microsatellite markers. Compare to many similar studies, this one is truly outstanding with respect to the study system - it is a subspecies of tropical butterfly, living at northern edge of its distribution, in geographically, ecologically and culturally fascinating region of southern tip of Florida, plus the adjacent islets of Florida keys. To add even more, the species seems to be closely tied to "pine rockland" habitat, which is, by itself, endangered, fragmented, and highly fragile.
So much being said, the manuscript suffers some small, but significant, mistakes regarding presentation. They are ameliorable, but the corrections must be carried out - otherwise, it loses much of its potential appeal.
1. you are comparing multiple populations/localities of the butterfly, which differ in size of habitat and level of fragmentation (line 117) but nowhere, except for the map in Figure 2, can a reader decipher which are the populations, or say sampling localities, how large and interconnected are the respective habitat patches, whether the species was there found abundantly or not, etc. All this information is a must, both for replicability and because it can give additional support for your conclusions. Ideally, depict all the populations to the map, and provide additional information about the sites - area, geographic coordinates, perhaps even some connectivity measures. You can, then, use some of the characteristics of the locations (area, connectivity) for correlations with the genetic indices from Table 2.
Plus, you are talking about "14 conservation properties" (again line 117), but only 7 populations are summarised in Table 2. Were they lumped - if so, how? - or do several properties cover some populations? If so, why to mention the 14 properties?
2. the paragraph starting at line 114 (Wolbachia). The whole justification for screening for Wolbachia is somehow naive. You are right that it may be male killing, sex-ratio distorting, etc., but there are populations with heavy Wolbachia loads not suffering anything apparent, and even observations that Wolbachia presence may maintain genetic cohesion of populations. I fully understand that this issue is rather messy at present, with new findings appearing on daily basis, and nobody willing to produce a cohesive review! To screen for the parasite is important, however, but not for an infestation control, but rather to collect basic data, plus, it is apparent that Wolbachia can distort genetic differentiation patterns - rendering some population genetic results meaningless, while providing a potential marker an sich. I recommend you this paper to illustrate the latter - you can cite it, if you wish, but I am not insisting: Suchackova Bartonova et al., 2021, Scientific Reports 11, 3019.
line 234: "Multiple analyses (including a principal coordinate analysis and a Bayesian 234 clustering analysis) confirm the general presence...."
Does it mean that you present only some of the multiple analyses? The paragraph needs re-writing, to adhere to more exact scientific style.
Finally, small remark. Consider supplying a photograph of the habitat, perhaps accompanying the map (Figure 2), or the portraits of the butterflies (Figure 3).
Author Response
Thank you for your kind and thorough review of our manuscript. Based on your comments, please find our detailed response. Please note that some additional changes/clarifications have been made in the text of the manuscript.
Reviewer 2 Comments:
- You are comparing multiple populations/localities of the butterfly, which differ in size of habitat and level of fragmentation (line 117) but nowhere, except for the map in Figure 2, can a reader decipher which are the populations, or say sampling localities, how large and interconnected are the respective habitat patches, whether the species was there found abundantly or not, etc. All this information is a must, both for replicability and because it can give additional support for your conclusions. Ideally, depict all the populations to the map, and provide additional information about the sites - area, geographic coordinates, perhaps even some connectivity measures. You can, then, use some of the characteristics of the locations (area, connectivity) for correlations with the genetic indices from Table 2. Plus, you are talking about "14 conservation properties" (again line 117), but only 7 populations are summarised in Table 2. Were they lumped - if so, how? - or do several properties cover some populations? If so, why to mention the 14 properties?
We added detail to the original map with callouts identifying each property from which tissue samples were collected along with their total habitat area in hectares. As most of the conservation lands properties are small and are surrounded by considerable urban matrix or other habitat types (or exist on individual islands). A total of 14 properties were surveyed but only 7 properties were found to have organisms – all of which were sampled for our analysis. No properties were lumped. Additional clarity is now included in the manuscript text. As the original purpose of the project was simply to compare population genetic structure across the network of occupied isolated pine rockland fragments, we did comprehensively sample to generate population abundance indices or map host coverage/patches. This information does not exist and was outside of our original scope. We simply sampled occupied properties broadly. Similarly, due to the configuration of each property and often broad sampling area, generating distances (connectivity) to other properties is challenging. A centroid could be generated but in many cases owing to the configuration of a site, this would exist outside of the conservation land unit. We plan to continue/expand this work in the future, applying a finer scale.
- The paragraph starting at line 114 (Wolbachia). The whole justification for screening for Wolbachia is somehow naive. You are right that it may be male killing, sex-ratio distorting, etc., but there are populations with heavy Wolbachia loads not suffering anything apparent, and even observations that Wolbachia presence may maintain genetic cohesion of populations. I fully understand that this issue is rather messy at present, with new findings appearing on daily basis, and nobody willing to produce a cohesive review! To screen for the parasite is important, however, but not for an infestation control, but rather to collect basic data, plus, it is apparent that Wolbachia can distort genetic differentiation patterns - rendering some population genetic results meaningless, while providing a potential marker an such. I recommend you this paper to illustrate the latter - you can cite it, if you wish, but I am not insisting: Suchackova Bartonova et al., 2021, Scientific Reports 11, 3019.
We describe our motivation for Wolbachia screening in the Introduction and provide an explanation of Wolbachia's varied impacts on Lepidoptera. We present our screening methods and confirm that no samples tested positive for Wolbachia (providing a confirmation of negative results, which are critical to the field of symbiont studies, but are rarely presented). Because we did not detect Wolbachia in any of the samples, it is not necessary to surmise population-level impacts or re-evaluate data due to Wolbachia influence (as there is none).
We have additionally added the following clarification into the manuscript text “Wolbachia may cause changes in population structure inferred by mitochondrial signatures (as described by Nice et al. in the Karner blue butterfly and by Bartonova et al. 2021 in Palaearctic blue butterflies ). As our study uses microsatellite markers (derived from nuclear DNA, not mitochondrial DNA), and did not detect any Wolbachia, we have increased confidence in our results of population structure.”
- line 234: "Multiple analyses (including a principal coordinate analysis and a Bayesian 234 clustering analysis) confirm the general presence...." Does it mean that you present only some of the multiple analyses? The paragraph needs re-writing, to adhere to more exact scientific style.
Both the PCoA analysis and the Bayesian clustering analysis are presented in their own figures (meaning, all analyses are presented). We will delete the word "including" and perhaps that would help?
- Finally, small remark. Consider supplying a photograph of the habitat, perhaps accompanying the map (Figure 2), or the portraits of the butterflies (Figure 3).
Done. Images of the habitat from two sampling locations have been added.